# Photo-Based Nanomedicines Using Polymeric Systems in the Field of Cancer Imaging and Therapy

**DOI:** 10.3390/biomedicines8120618

**Published:** 2020-12-16

**Authors:** Patihul Husni, Yuseon Shin, Jae Chang Kim, Kioh Kang, Eun Seong Lee, Yu Seok Youn, Taofik Rusdiana, Kyung Taek Oh

**Affiliations:** 1Department of Global Innovative Drugs, College of Pharmacy, Chung-Ang University, 221 Heukseok dong, Dongjak-gu, Seoul 06974, Korea; patihul.husni@unpad.ac.id (P.H.); sus9417@cau.ac.kr (Y.S.); wofl2018@cau.ac.kr (J.C.K.); rldh4122@cau.ac.kr (K.K.); 2Department of Biotechnology, The Catholic University of Korea, 43 Jibong-ro, Bucheon-si 14662, Gyeonggi-do, Korea; eslee@catholic.ac.kr; 3School of Pharmacy, Sungkyunkwan University, 2066 Seobu-ro, Jangan-gu, Suwon 16419, Gyeonggi-do, Korea; ysyoun@skku.edu; 4Department of Pharmaceutics and Pharmaceutical Technology, Faculty of Pharmacy, Universitas Padjadjaran, Jatinangor 45363, Indonesia; t.rusdiana@unpad.ac.id

**Keywords:** light, optical imaging, phototherapy, nanomedicine, photodynamic therapy, photothermal therapy

## Abstract

The use of photo-based nanomedicine in imaging and therapy has grown rapidly. The property of light in converting its energy into different forms has been exploited in the fields of optical imaging (OI) and phototherapy (PT) for diagnostic and therapeutic applications. The development of nanotechnology offers numerous advantages to overcome the challenges of OI and PT. Accordingly, in this review, we shed light on common photosensitive agents (PSAs) used in OI and PT; these include fluorescent and bioluminescent PSAs for OI or PT agents for photodynamic therapy (PDT) and photothermal therapy (PTT). We also describe photo-based nanotechnology systems that can be used in photo-based diagnostics and therapies by using various polymeric systems.

## 1. Introduction

For the past several decades, many studies focused on eradicating cancer, one of the most dreaded diseases, have been reported that use new treatment modalities as well as conventional treatments, such as surgery, chemotherapy, and radiotherapy [1]. Owing to the limitations of conventional treatments, such as their side effects (e.g., functional, physical, and psychological impairment, the occurrence of multidrug resistance, inadequate selectivity to target location, and low efficacy (e.g., chemotherapy)), many researchers are devoted to discovering alternative, less-invasive approaches to combat cancer [2,3,4]. Recently, photo-based treatment in cancers has attracted more interest because of its great potential in clinical cancer therapy [5,6,7].

Light has been used to treat vitamin D deficiency, autoimmune diseases, neonatal jaundice, and skin-related diseases [8,9]. The energy from light, a form of electromagnetic radiation that comprises photons, can be converted into different forms, such as heat, chemical energy, and acoustic waves [10,11,12]. Notably, absorbed light can induce changes in photosensitive agents (PSAs) administered in the body. PSAs can then react as photophysical or photochemical molecules for diagnostic or therapeutic purposes [13,14,15]. The rapid development of nanotechnology has enabled it to serve as a good prospect for the development of photosensitive nanomedicines for the above purposes [16,17,18].

In recent years, nanomedicines with good potential for use in phototherapy (PT) have become popular research topics. The advantages of nanotechnology, such as targetability, surface modification, environmental responsiveness, and economic preparation, could promote the development of innovative drug delivery systems for cancer diagnosis and therapy at a molecular level, ultimately improving treatment efficacy and reducing side effects [19,20,21,22]. Based on photosensitive applications for imaging and therapy, nanomedicines incorporated with PSAs have many advantages, including high PSA loading capacity in nanoparticles (NPs), protection of PSAs from degradation and photo-bleaching, extended circulation times, and optimized distribution in vivo [23,24,25]. In particular, the lack of selectivity to target locations can be overcome with the use of nanotechnology, which would lead to decreased damage to healthy tissues and improved PT efficacy in tumors via the enhanced permeability and retention (EPR) effect [26,27,28,29]. Following intravenous (IV) administration, the PSA-incorporated NPs can be activated by light, inducing fluorescence for imaging or radical molecules for PDT, or elevating temperature for photothermal therapy (PTT) (Figure 1). The most widely studied drug delivery systems based on nanoparticle technology are liposomes, polymers, and solid inorganic NPs [16]. Among them, we focus on polymeric-based nanomedicines. In this review, we address the benefits and risks of using light, imaging agents, PT, and photosensitive nanomedicines derived using polymers of different architectures.

## 2. Benefits and Risks of Photo-Based Imaging and Therapy

Nowadays, light is widely employed in disease applications, including cancers, for therapy and imaging. However, the risks as well as benefits of using light for imaging and treatment should be carefully examined. The benefits and risks of photo-based imaging and therapy are presented in Table 1.

PT can be established by controlling time, treatment sites, efficacy at the irradiation area, duration, and the power of light [4,13]. When PSA nanomedicines are used in PT, the activation of non-toxic PSAs using local light irradiation results in the selective killing of target cells (e.g., cancer cells), with only minor damages occurring in normal tissues. The careful design of phototherapeutic nanomedicines and well-controlled light irradiation in the location of lesions (e.g., tumor tissues) maximize their efficacy because of the dual selectivity of PT [37,38,39]. Unlike surgery, minimally invasive techniques requiring a small insertion of an instrument into a body cavity can be conducted with a flexible optical fiber-bundle device that has a diameter of ~200–300 µm; this is very suitable for imaging deep within tissues or delivering light for PDT [40,41,42,43]. Non-invasive photo-based imaging and therapy within tissues or the body can also be achieved with near-infrared (NIR) light in the range of 650–900 nm, thereby enabling deep tissue propagation with low attenuation [44,45,46]. According to clinical demands, treatment can be easily adjusted by adapting the procedure of photo-based imaging and therapy [13].

Photo-based nanomedicines should be scrutinized for their risks. Light damages, such as photic injury, photochemical injury, and photomechanical injury, can be induced according to irradiation power density, irradiation time, spot size, wavelength, and manner of exposure (e.g., irradiation frequency) [33,47,48]. Photothermal damage, which involves the heating of tissues by the absorbed energy, is the most common type of photic injury. At the cellular and molecular levels, an increase in temperature leads to protein denaturation, molecular tertiary structure loss, and membrane fluidization. When photochemical injury is caused by long durations and high energy (or low wavelength) of light exposure, free radicals generated under these conditions can interact with endogenous chromophores and oxidize proteins and cell membrane lipids, causing painful eye injury, premature skin aging, skin burning, or skin cancer [49,50]. Furthermore, exposure to high-energy light (megawatts or terawatts/cm^2^), even if the duration of irradiation is short (nanoseconds to picoseconds), can cause photomechanical injury via compressive or tensile forces to tissues [13,33]. When light is employed for optical imaging (OI) and therapy, the potential of phototoxicity, which can be caused by drugs and essential oils that diffuse into the skin or eyes, should be considered [13,33,51,52,53]. In particular, high exposure to ultraviolet (UV) light increases the risk of skin cancer development. Frequent treatment with light can suppress the immune system, leaving the body vulnerable to diseases, infections, and skin cancers [34,54]. In fact, the more treatment one undergoes and the fairer the skin, the higher the risk of skin cancer [35,36]. Based on the benefits and risks described above, imaging and therapy using light should be carefully considered.

## 3. Optical Imaging

As OI is the most universal visualization technique, it is extensively used in many research areas [55,56,57]. Non-ionizing radiation ranging from UV to NIR light has been used in OI [58,59,60]. OI is associated with lower risks in patients and a faster analysis process and serves as a sensitive method for visualizing biological processes in vivo. In addition, the properties of OI can enable long-term or repetitive observation of disease progression [61,62]. One of the fundamental advantages of OI is the accessibility it provides to interactions between light and tissue and the corresponding photophysical and photochemical processes at the molecular level. In the OI process, imaging PSAs can produce detectable and targeted signals after injection into the body. PSAs can then be detected with high sensitivity, high toxicity, low toxicity, good solubility in aqueous media, high fluorescence quantum yield, high resolution, and prolonged fluorescence lifetime. Table 2 shows some of the widely used PSAs in OI.

Fluorescence imaging is based on the illumination of a target tissue with a specific wavelength or wavelength range (e.g., from UV to NIR) from a light source. The types of fluorescence compounds used for imaging include fluorescent dyes, quantum dots, and fluorescent proteins [67,73,74]. The interaction between the photons and PSAs results in the excitation of the PSAs. This excited light penetrates the tissue layers to reach the PSAs and, consequently, is partially reflected and scattered. PSAs emit photons with specific wavelengths on their subsequent return to the basal energetic state; these photons are then captured with an array of detectors for imaging [75,76,77].

In bioluminescence imaging, luciferin and luciferase are used as the light-emitting molecule and oxidizing enzyme, respectively. The oxidation of luciferin by luciferase in the presence of co-factors, such as adenosine triphosphate (ATP) and magnesium, results in the release of photons as the substrate returns from its electronically excited state to its ground state, ultimately emitting light with a broad emission spectrum (red and far-red emission) for imaging [78,79,80]. Previously, bioluminescence was employed to detect metastatic cancer, imaging protein interactions, and signaling pathways. For example, Stollfuss et al. used bioluminescence to image tumorigenesis and metastasis in a mouse model [81,82,83,84]. Further, Stowe et al. applied bioluminescence to the monitoring of tumor burden and cell tracking of chimeric antigen receptor (CAR) T cell therapy within a single animal model [85,86]. Bioluminescence imaging can also be used to monitor infectious disease models. In fact, Luker et al. demonstrated the application of bioluminescence imaging to provide information not only on the interaction among host and pathogen of luciferase-expressing viruses, bacteria, and fungi but also on the real-time response to antiviral or antibiotic treatments [87].

## 4. Phototherapy

PT involving the irradiation of light for the treatment of diseases such as cancer can be broadly classified into two categories: PDT and PTT [13,88]. These commonly used fixed-wavelength light is used to activate the administered PSAs. Treatment using PT can be administrated as a stand-alone therapy or be combined with chemotherapy agents to achieve synergistic effects [69,88,89,90,91,92,93,94]. PDT based on noninvasive photochemistry utilizes the generation of highly reactive singlet oxygen, the excited state of molecular oxygen (^1^O_2_), which can destroy the target cells via oxidative stress [95,96,97]. The irradiation process in PDT involves three components: light, PSAs, and molecular oxygen. As the PSAs are irradiated by light of an appropriate wavelength and power to absorb a photon and excite an electron, they are promoted to an excited singlet state from the ground singlet state (electron-paired). The excited singlet state of lower-energy orbital moves to a lower-energy excited triplet state (electrons unpaired) accompanied by fluorescence emission, loss of energy as heat, or other photophysical energy. The excited PSAs also produces reactive oxygen species (ROS) via a direct reaction with other biomolecules or ground state oxygen [13,88,98]. The ROS generated in the PDT can induce damage to the target tissue as a form of cell necrosis by rupturing the cell membrane and causing cell apoptosis via the activation of several signal pathways [95,96,97].

In PTT, light can increase the temperature in PSAs exposed to specific wavelengths of visible (Vis) or NIR light [99,100,101]. Similarly, irradiation of light to a target location after PSA administration excites the PSAs, which undergo internal conversion to the ground state. The conversion of electrons from the excited state to the ground state results in the emission of energy in the form of heat and increases the surrounding temperature [102,103,104]. The resulting hyperthermia can cause irreversible cell damage at 42–46 °C if the duration of treatment is more than 10 min. The higher the temperature provided, the shorter the treatment time required [13].

PSAs for PDT require a high molar extinction coefficient as well as high-energy and long-lived triplet states to induce singlet oxygen with a high quantum yield. However, PSAs for PTT assume a high molar extinction coefficient, a very low quantum yield of fluorescence, and a short-lived and low-energy triplet state (pico-second range) [13,105,106,107,108]. The PSAs commonly used and investigated in PT are presented in Table 3.

## 5. Photosensitive Nanomedicines

Photosensitive nanomedicines based on nanotechnologies have been studied to better elucidate their application in photo-based diagnostics and therapies. Nanomedicines with photosensitivity include NPs that were PSAs, photo-triggered carriers loading drugs, and nano-sized carriers containing PSAs [130,131,132].

First, some NPs such as TiO_2_, ZnO, and fullerene for PDT can act as PSAs through the generation of singlet oxygen. Metallic NPs such as gold NPs act as PSAs for PTT induced by photothermal effects (i.e., plasmonic PTT). Gold NPs can absorb light (photonic energy) effectively and convert it efficiently into heat energy [102,103,104]. The thermal energy created by gold NPs is dependent on the interactions between light and NP, which occur via a surface plasmon resonance effect [102,104,133].

Photo-triggered systems include photosensitive polymers and anticancer drugs. Generally, light irradiation can remotely affect the photo-responsive carriers in cancer cells [134,135,136]. First, the optical signal is captured by the photochromic molecules (chromophores), which convert photo-irradiation into a chemical signal through a photoreaction, ultimately causing the release of drugs by the change in carrier structures. The photo-triggered system utilizes photo-responsive chemistry, such as photoisomerization using azobenzene (AZO) [137], spiropyran (SP), and dithienylethene (DTE) [138]; photo-induced rearrangement using 2-diazo-1,2-naphthoquinone (DNQ) [139]; photo-based cleavage using o-nitrobenzyl ester [140], coumarinyl ester [141], and pyrenylmethyl ester [142]; and photo-induced energy conversion using AZO derivatives [143].

Nano-sized systems containing PSAs have been demonstrated to exploit the enhancements in PSA delivery, such as the high concentration of PSAs in NPs, protection from degradation and photobleaching, prolonged circulation times, and optimized distribution in vivo [24,144,145,146,147]. However, free PSAs tend to be completely cleared upon administration (in vivo) and are less effective [13]. Among the nano-sized carriers, polymeric drug systems, which possess several favorable properties, have been examined to elucidate their potential application in photosensitive nanomedicines. These favorable properties include easy fabrication, tailor-made design, versatile functionality, responsiveness to environmental stimuli, and high drug loading capacity [148,149,150,151,152,153,154]. In the next section, nanomedicines incorporating PSAs prepared using polymers of different architectures are discussed.

## 6. Polymer-Based Photosensitive Nanomedicine

Polymers with different architectures, such as linear, branched, and crosslinked polymers, have been designed as PSA delivery systems for the treatment of cancers [37,131,155]. These polymers exhibit different physicochemical properties, such as hydrodynamic properties, melt rheology, and mechanical performance, depending on their structures [156]. Photo-based nanomedicines fabricated with polymers have been prepared by loading PSAs in the hydrophobic block or conjugating PSAs to the polymer backbone. After IV injection, PSAs incorporating nanomedicines can circulate systemically and accumulate in tumor tissues via the EPR effect [157]. The nanomedicines delivered to the target sites enable the capture of an image or reveal the cytotoxicity induced by irradiation because of the adequate light for PSAs. Various polymers with different structures and PSAs have been used to design photo-based nanomedicines (Figure 2).

### 6.1. Photosensitive Nanomedicine Using Linear Polymers

For linear polymers, several block copolymers, such as AB-, ABA, and ABC-type, are employed as PSA carriers. For the desired purposes of photosensitive nanomedicines, the block copolymers are combined with a hydrophobic block to incorporate a pool of water-insoluble agents, a polyelectrolyte to achieve a conjugating backbone of PSAs or stimuli responsiveness, and/or a hydrophilic block that provides stability and enables extended circulation throughout the body [1,111,130,131].

Oh’s group developed several block copolymers that exhibited structural changes through a response to an acidic condition of the tumor extracellular pH (pH_ex_) or endosomal pH (pH_en_). Poly (aspartic acid-graft-imidazole)-poly(ethylene glycol) (PAIM-PEG) was synthesized as a pH-sensitive nanocarrier of the photosensitizer, indole-3-acetic acid (IAA), for the treatment of skin cancer (Figure 3). IAA-loaded micelles (ILMs) resulted in the formation of spherical particles (ca. 140 nm) at pH 7.4, pH-dependent IAA release, and cytotoxicity due to micelle disintegration at acidic pH. Notably, when ILMs were administered as treatment with nontoxic Vis light at a wavelength of 480 nm, synergistic pH-dependent cell damage was observed under Vis light irradiation in both in vitro and in vivo models using the B16F10 melanoma cell line; this was confirmed via ROS production at an acidic pH of 6.5 [158]. These researchers also prepared an on-demand pH-sensitive nanocluster (NC) system with gold nanorods and doxorubicin (Dox) using PAIM-PEG (Figure 4). The NC system showed less systemic toxicity at pH 7.4 due to the formation of a robust nano-assembly and enhanced antitumor efficacy via the synergic effect of increased Dox release at pH_ex_ and pH_en_, as well as a gold nanorod photothermal effect with locally applied NIR light [131]. A stable polyelectrolyte nanoparticle composed of PEG-poly(l-lysine)-poly(lactic acid) (PEG-PLL-PLA) for PDT was also constructed. The photosensitizer, Chlorin e6 (Ce6), and a pH-responsive 2,3-dimethyl maleic anhydride (DMA) moiety were conjugated to the lysine residue in PEG-PLL-PLA, resulting in the PEG-PLL(-g-Ce6, DMA)-PLA triblock copolymer. The micellar systems were found to accumulate at the tumor site and substantially inhibit the growth of KB cell line [111].

For PDT, Wang et al. [159] designed a novel photosensitizer carrier with an oxygen self-compensating ability using synthesized PEG-poly(acrylic acid)-polystyrene (PEG-PAA-PS) followed by chemical conjugation of hemoglobin (Hb) (Figure 5). Zinc phthalocyanine (ZnPC), a second-generation photosensitizer, was encapsulated in the Hb-conjugated PEG-PAA-PS micelles. The micelles could generate more ^1^O_2_ in the presence of Hb and induce more significant photocytotoxicity on HeLa cells than on those without Hb. Li et al. [160] reported a convenient and universal approach to spatiotemporally control the chemodrug release via a PDT-mediated alteration of the tumor microenvironment. Briefly, the micelles were formed by an arylboronic ester (BE)-modified PEG-2-((((4-(4,4,5,5-tetramethyl-1,3,2-dioxaborolan-2-yl)benzyl)oxy)carbonyl)amino) ethyl methacrylate, which was used to encapsulate Dox and hematoporphyrin (Hp). The Dox/Hp co-encapsulated micelles were stable under normal physiological conditions and displayed a uniform size distribution (∼100 nm). Extensive ROS is generated from Hp at the tumor sites under tumor-specific light irradiation, thereby rapidly dissociating the micelles and selectively releasing the Dox as a consequence of the ROS-mediated cleavage of the hydrophobic BE moieties on the polymers; this results in synergistic anti-cancer effects of the Dox-mediated chemotherapy and the Hp-mediated PDT.

For PTT, Pan et al. [161] prepared PEG-poly(L-aspartic acid sodium salt)10 (PLD) micelles, including heptamethine cyanine (IR825) with polarity-sensitive fluorescence characteristics. These micelles were not only beneficial for in vitro imaging (Ex: 552 nm, Em: ~610 nm) but also for in vivo NIR fluorescence imaging-guided PTT (Ex: 780 nm, Em: 830 nm). In addition, an in vivo study revealed that PEG-PLD (IR825) micelles possess promising tumor ablation ability during PTT. Guo et al. [162] designed and synthesized a donor–acceptor structured porphyrin-containing conjugated polymer, poly([(5,15-diethynyl-10,20-bis(3,5-bis(octyloxy)phenyl)porphyrin]zinc-alt-(2,1,3-benzothiadiazole) (PorCP), for efficient PTT in vitro and in vivo. The porphyrin in the polymer acted as a backbone and displayed an absorption peak at 799 nm. The NPs formed by PorCP displayed favorable nonradiative decay, good photostability, a high extinction coefficient at 800 nm based on the molar concentration of porphyrin, and a remarkable photothermal conversion efficiency (63.8%). Yang et al. [163] used an organic conductive polymer, polypyrrole (PPy), for photothermal ablation of cancer in vitro and in vivo at ultra-low laser power density (Figure 6). The NPs formed by polypyrrole showed high stability and little cytotoxicity in physiological environments and a laser power-dependent cancer cell ablation effect. The result of intratumoral injection of NPs revealed excellent tumor treatment efficacy using an ultra-low power of NIR laser irradiation at 0.25 W/cm^2^ (75 J/cm^2^). Further, 100% tumor elimination was achieved without any marked toxic side effects post-treatment.

### 6.2. Photosensitive Nanomedicine Using Branched Polymers

Branched polymers, such as star-shaped and dendron/dendrimers, have been gaining attention due to their potential for application in photo-based nanomedicines. In fact, several star-shaped polymers have been used as photo-responsive delivery systems. Qu et al. [164] developed a star-shaped micellar system by mixing a photoinitiated crosslinking amphiphilic copolymer containing cinnamyl groups with a phenylboronic acid (PBA)-functionalized redox-sensitive amphiphilic copolymer (Figure 7). The end groups of the hydrophilic segments were decorated with PBA ligands to provide active targeting ability. A redox response was triggered by the disulfide bonds in the micellar matrix to achieve rapid intracellular release of drugs. The results of the in vivo antitumor effect on H22-bearing BALB/c mice showed that the micelles had high therapeutic efficacy against solid tumors, with minimal side effects against normal tissues. Dai et al. [165] synthesized star-shaped porphyrin-cored PLA-poly(gluconamidoethyl methacrylate) for targeted PDT. Under irradiation, the copolymer exhibited efficient singlet oxygen generation and displayed high-fluorescence quantum yields. When a longer irradiation time was applied, more BEL-7402 cancer cells were found to die.

For dual delivery systems, Zhang et al. [166] reported the high tumor-targeting and anticancer effects of Dox-loaded photosensitizer-core pH-responsive copolymer nanocarrier prepared from a four-armed star-shaped copolymer, [methoxy-poly(ethylene glycol)-poly(2-(*N*,*N*-diethylamino)ethyl methacrylate)-poly(ε-caprolactone)]_4_-zinc β-tetra-(4-carboxyl benzyloxyl)phthalocyanine (PDCZP). The nanocarriers loaded with zinc phthalocyanine (ZnPc) had a long emission wavelength (max. 677 nm) and could generate singlet oxygen (^1^O_2_). The Dox-loaded nanocarriers showed improved in vitro and in vivo anticancer effects under irradiation, with rapid Dox release from nanocarriers in acidic media. Gangopadhyay et al. [167] formulated single-component fluorescent organic polymeric NPs using a star-shaped four-arm PEG containing the chromophore, coumarin, which acted as a photosensitizer and photo-trigger molecule. By itself, it displayed PDT and enabled the simultaneous release of the chemotherapeutic drug, chlorambucil, upon irradiation with light. The anticancer drug was released by the coumarin chromophore in a photo-controlled manner, and the coumarin generated singlet oxygen (^1^O_2_) upon irradiation with UV/Vis light (≥365 nm). In vitro study results using the HeLa cell line revealed a reduction in cell viability of up to ~5% when a combined treatment of PDT and chemotherapy was administered. An’s group constructed multi-stimuli-responsive NPs through the co-assembly of a three-arm star quaterpolymer with cypate and paclitaxel as a photothermal cyanine dye and chemotherapeutic compound, respectively (Figure 8). The NPs enhanced the photothermal effect and preferred NIR-light-triggered drug release in the acidic environment as well as lysosomal disruption-mediated intracellular drug translocation. The NPs also exhibited enhanced cellular uptake and tumor accumulation [168].

To utilize dendrimers, Kojima et al. [169] prepared two PEG-attached dendrimers derived from poly(amido amine) (PAMAM) and poly(propylene imine) (PPI) dendrimers to form nanocapsules of the photosensitizers, rose bengal (RB), and protoporphyrin IX (PpIX) (Figure 9), for PDT. Compared to free PpIX, the complex of PpIX with PEG-PPI exhibited efficient cytotoxicity. Further, Taratula et al. [170] encapsulated silicon naphthalocyanine (SiNc) into the hydrophobic interior of a generation 5 PPI dendrimer following surface modification with PEG. The NPs showed robust heat generation capability (ΔT = 40 °C) and efficiently produced ROS under NIR irradiation (785 nm, 1.3 W cm^−2^) without releasing SiNc from the nanoplatform. With NIR irradiation, the PT mediated by SiNc efficiently destroyed chemotherapy-resistant ovarian cancer cells and prevented cancer recurrence. Yuan et al. [171] developed a light-and-pH dually responsive amphiphilic dendrimer-star copolymer, poly(ε-caprolactone)-block-poly(methacrylic acid-co-spiropyran methacrylate), for merocyanine PSA. The SP groups exhibited light- and pH-dually responsive properties through UV light irradiation and altered the pH values of the micelle solutions. Upon UV light irradiation or at low pH, the hydrophobic SP isomerized to hydrophilic merocyanine or merocyanine H^+^. The copolymer micelles possessed good biocompatibility and were thus employed as the drug delivery system for the controlled release of the anticancer drug Dox.

### 6.3. Photosensitive Nanomedicine Using Crosslinked Polymers

Crosslinked polymers in biomedical nanoplatforms possess many advantages, such as robustness as well as their role as a protector of the loading agents and an enhancer of cellular uptake [172,173,174]. Photosensitive nanomedicines constructed with crosslinked polymers have been utilized for imaging and PT [172]. In fact, Tang et al. [175] constructed NPs comprising photo-cross-linkable semiconductor polymer dots (Pdots) doped with the photosensitizer Ce6. The photoreactive oxetane groups were attached to the side chains of the semiconductor polymer, do-PFDTBT, which was polymerized with 9,9-Di-{6-[(3-methyloxetan-3-yl)methoxy]hexyl)-2,7-di[boronicacid bis(pinacol) ester]-fluorene and 4,7-bis(5-bromo-4-hexylthiophen-2-yl)benzo[*c*][1,2,5]thiadiazole. Following the photo-cross-linking reaction, the Ce6-doped Pdots formed an interpenetrated structure to prevent the leaching of Ce6 from the Pdot matrix. The Ce6-doped Pdots (∼10 μg/mL) effectively suppressed the cancer cells under low doses of light irradiation (∼60 J/cm^2^). Li et al. [176] synthesized a crosslinked nanogel for in vivo imaging of caspase activity. The nanogel was prepared with gold NPs that were modified with a vinyl group and Cy5-linked CPADVEDK peptides (Figure 10). However, the shell of such nanogels can be degraded under acidic conditions, and caspase-3 or -7 can further cleave the peptide to release Cy5. By employing in vivo fluorescence to observe the tumor, the nanogel-injected mice were found to have the highest fluorescence signal at the tumor site, ultimately demonstrating the in vivo activation of the nanogels.

Chambre et al. [177] prepared porphyrin-cross-linked nanogels via the self-assembly and in situ cross-linking of thermoresponsive copolymers, poly[(PEG-based methacrylate-co-azide-containing methacrylate (poly(PEGMEMA-*co*-AHMA)). To synthesize the nanogels, PEG-methacrylate-based copolymers containing reactive azide groups as side chains were assembled into nanosized aggregates and cross-linked with a tetra-alkynyl Zn porphyrin using the copper-catalyzed azide−alkyne cycloaddition reaction under surfactant-free conditions (Figure 11). The nanogels containing porphyrin retained their singlet oxygen generation ability of the porphyrin core and could induce a temperature increase upon irradiation at 635 nm. Further, the porphyrin-loaded nanogels could induce anticancer effects owing to their accumulation in the cytoplasm of cells when illuminated at short and long wavelengths. Ji et al. [178] hybridized graphene oxide (GO) into poly(*N*-isopropylacrylamide) (PNIPAM) nanogels, which led to good stability and high photothermal effects. The hybrid nanogels accelerated drug release under conditions mimicking the acidic solid tumor and intracellular microenvironments and were further enhanced via remote photothermal treatment.

## 7. Conclusions

Photo-based nanomedicines have been successfully used in imaging and therapy applications. However, the risks and benefits of using light should be considered during its selection for use in diagnostics and therapies. Owing to remarkable advances in nanotechnology, photosensitive nanomedicines can be designed and prepared for applications in photo-based diagnostics and therapies such as photo-triggered systems, NPs containing PSAs, and NPs that are themselves PSAs. Photosensitive nanomedicines are fabricated with several polymers with different architectures, such as linear, branched, and crosslinked structures. In this review, we sought to describe the potential of using photosensitive nanomedicine for diagnosis and therapy. If a photo-based nanomedicine system is successfully developed, several challenges will arise prior to its use in the clinic, such as its nontoxicity, targeting properties, and reduced unwanted cargo release.

## Figures and Tables

**Figure 1 biomedicines-08-00618-f001:**
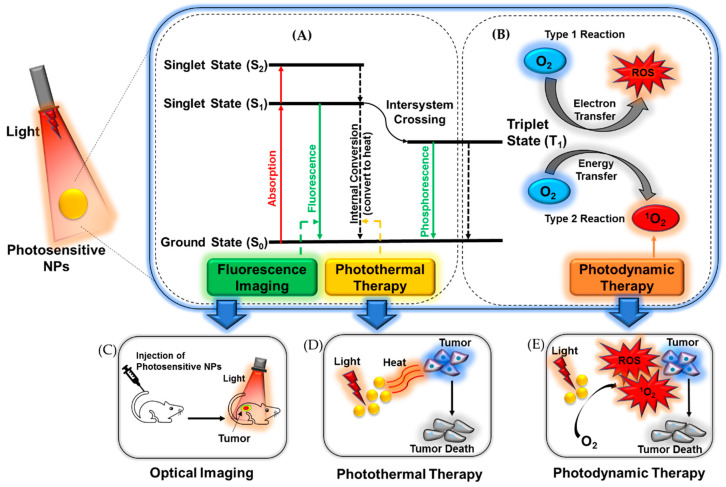
Illustration of the basic mechanism used in (**A**) fluorescence imaging and PTT, and (**B**) PDT and use of photosensitive nanomedicine in (**C**) OI, (**D**) PTT, and (**E**) PDT. NPs: nanoparticles; ROS: reactive oxygen species; PTT: photothermal therapy; PDT: photodynamic therapy; OI: optical imaging.

**Figure 2 biomedicines-08-00618-f002:**
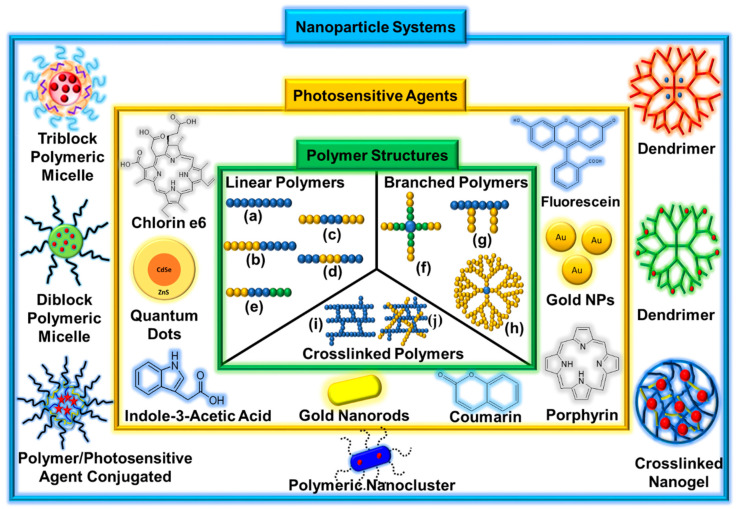
Schematic illustration of polymer structures, photosensitive agents, and nanoparticle systems. (**a**) Homopolymer, (**b**) AB-type diblock copolymer, (**c**) ABA-type triblock copolymer, (**d**) BAB-type triblock copolymer, (**e**) ABC-type triblock copolymer, (**f**) star-shaped block copolymer, (**g**) graft copolymer, (**h**) dendrimer, (**i**) polymer networks, and (**j**) interpenetrating polymer networks (IPN).

**Figure 3 biomedicines-08-00618-f003:**
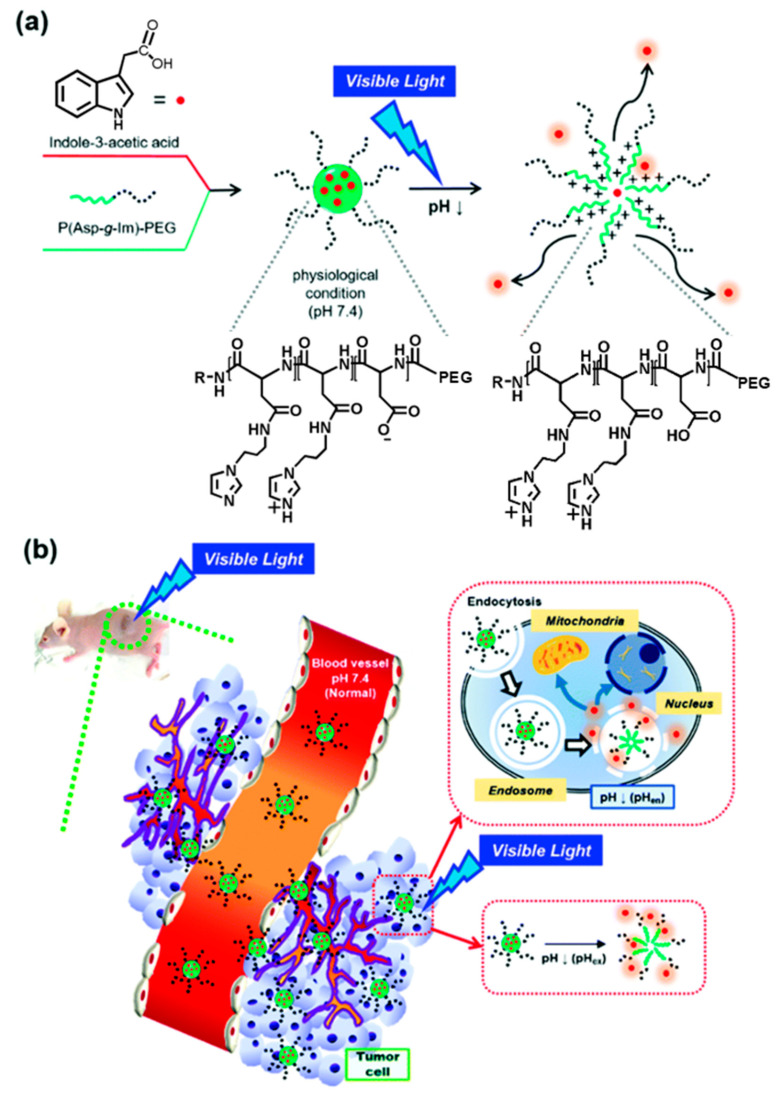
Schematic concepts for an improved photodynamic therapeutic effect using IAA-loaded micelles (ILMs): (**a**) hypothetical mechanism for the behavior of ILMs and (**b**) proposed performance of ILMs in vivo. Reproduced with permission from Sim, T.; Lim, C.; Hoang, N.H.; Kim, J.E.; Lee, E.S.; Youn, Y.S.; Oh, K.T. Synergistic photodynamic therapeutic effect of indole-3-acetic acid using a pH-sensitive nano-carrier based on poly(aspartic acid-graft-imidazole)-poly(ethylene glycol). *J. Mater. Chem. B*
**2017** [158].

**Figure 4 biomedicines-08-00618-f004:**
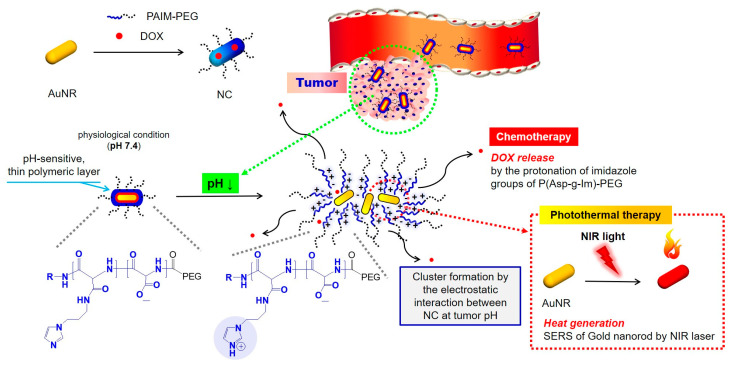
Graphical abstract of pH-sensitive NC system with gold nanorods and Dox using PAIM-PEG. AuNR: gold nanorods; NC: nanocluster; Dox: doxorubicin; PAIM-PEG: poly (aspartic acid-graft-imidazole)-poly(ethylene glycol) [131].

**Figure 5 biomedicines-08-00618-f005:**
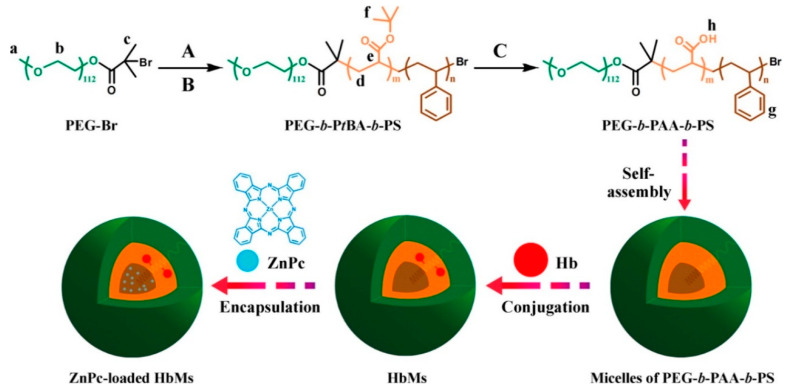
Synthesis of ZnPc-loaded hemoglobin conjugated micelles (HbMs): (**A**) *tert*-butyl acrylate (*t*BA), copper(I) bromide (CuBr), *N*,*N*,*N*′,*N*″,*N*″-pentamethyl diethylene triamine (PMDETA), 50 °C; (**B**) styrene, CuBr, PMDETA, chlorobenzene, 110 °C; (**C**) trifluoroacetic acid, chloroform, 25 °C. PEG: poly(ethylene glycol); P*t*BA: poly(*tert*-butyl acrylate); PS: polystyrene; PAA: poly(acrylic acid); Hb: hemoglobin; HbMs: hemoglobin-conjugated micelles; ZnPc: zinc phthalocyanine. Reproduced with permission from Wang, S.; Yuan, F.; Chen, K.; Chen, G.; Tu, K.; Wang, H.; Wang, L. Synthesis of hemoglobin conjugated polymeric micelle: A ZnPc carrier with oxygen self-compensating ability for photodynamic therapy. *Biomacromolecules*
**2015** [159].

**Figure 6 biomedicines-08-00618-f006:**
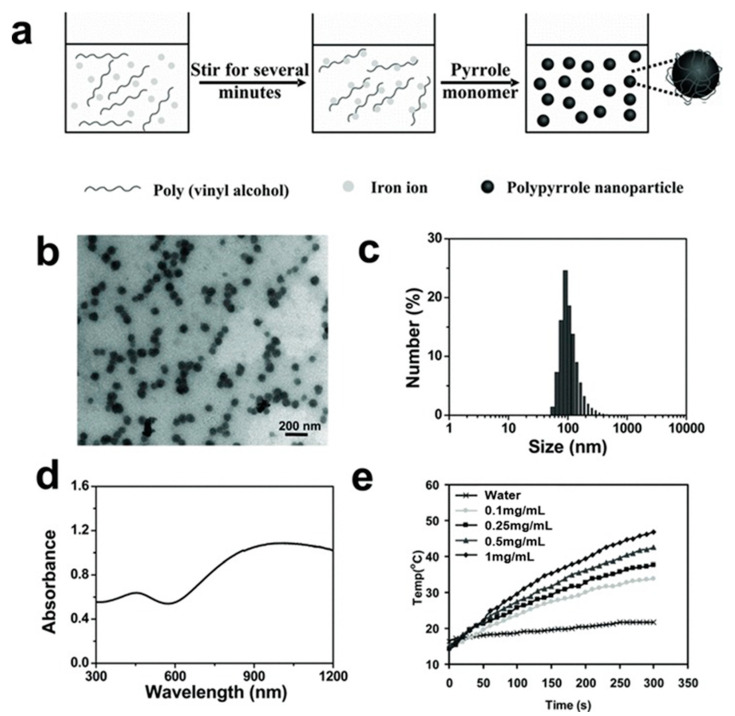
(**a**) Schematic representation of synthesis of poly(vinyl alcohol) (PVA). (**b**) Transmission electron microscopy (TEM) image of PPy NPs. (**c**) Dynamic light scattering (DLS) data of PPy NPs. (**d**) UV-Vis-NIR absorbance spectrum of a PPy solution (inset: photo of PPy solutions in water, saline, and fetal bovine serum). (**e**) Heating curves of water and various PPy concentration (0.1, 0.25, 0.5, and 1 mg/mL), 808 nm laser irradiation, and power density of 0.5 W/cm^2^. PPy: polypyrrole; NPs: nanoparticles. Reproduced with permission from Yang, K.; Xu, H.; Cheng, L.; Sun, C.; Wang, J.; Liu, Z. In vitro and in vivo near-infrared photothermal therapy of cancer using polypyrrole organic nanoparticles. *Adv. Mater.*
**2012** [163].

**Figure 7 biomedicines-08-00618-f007:**
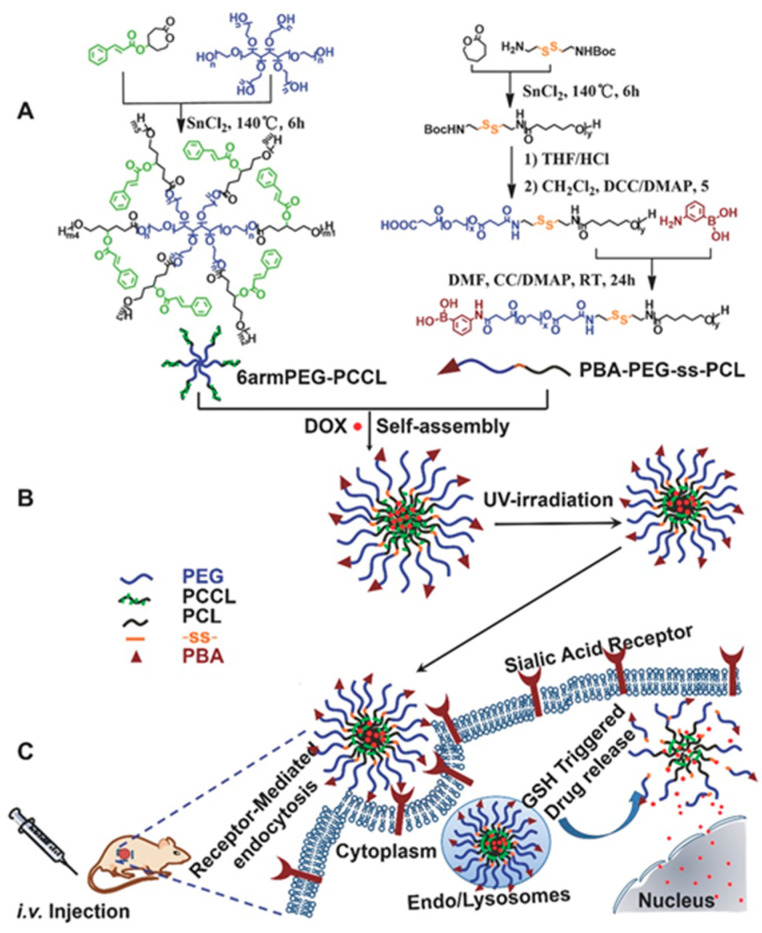
(**A**) Synthetic route and chemical structure of 6arm PEG-PCCL and PBA-PEG-ss-PCL copolymers. (**B**) Schematic illustration of DOX-loaded star-shaped mixed micelle formation and photo-crosslinking under 365 nm UV light irradiation. (**C**) Phenylboronic acid-mediated endocytosis of these multifunctional micelles and subsequently GSH-triggered intracellular drug release. Reproduced with permission from Qu, Q.; Wang, Y.; Zhang, L.; Zhang, X.; Zhou, S. A nanoplatform with precise control over release of cargo for enhanced cancer therapy. *Small*
**2016** [164].

**Figure 8 biomedicines-08-00618-f008:**
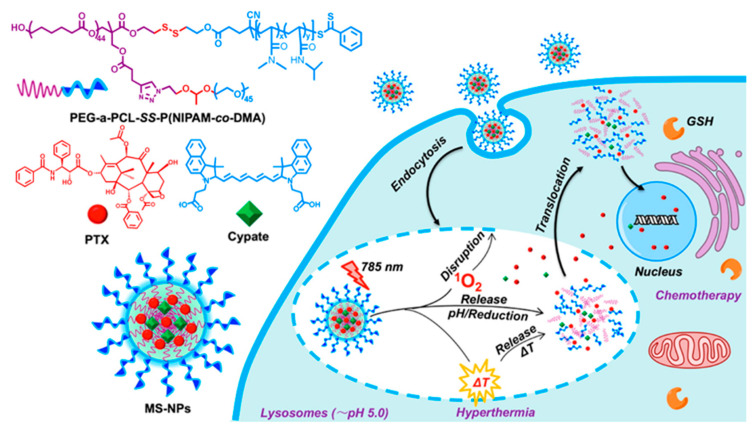
Schematic illustration of NIR light/pH/reduction–responsive nanoparticles consisting of PEG-*a*-PCL-SS-P(NIPAM-*co*-DMA) star quaterpolymer for precise cancer therapy with synergistic effects. PEG: poly(ethylene glycol); PCL: poly(ε-caprolactone); SS: disulfide; P(NIPAM): poly(N-isopropylacrylamide; DMA: *N*,*N*-dimethylacrylamide. GSH: glutathione; MS-NPs: multi-stimuli-responsive nanoparticles; PTX: paclitaxel. Reproduced with permission from An, X.; Zhu, A.; Luo, H.; Ke, H.; Chen, H.; Zhao, Y. Rational design of multi-stimuli-responsive nanoparticles for precise cancer therapy. *ACS Nano*
**2016** [168].

**Figure 9 biomedicines-08-00618-f009:**
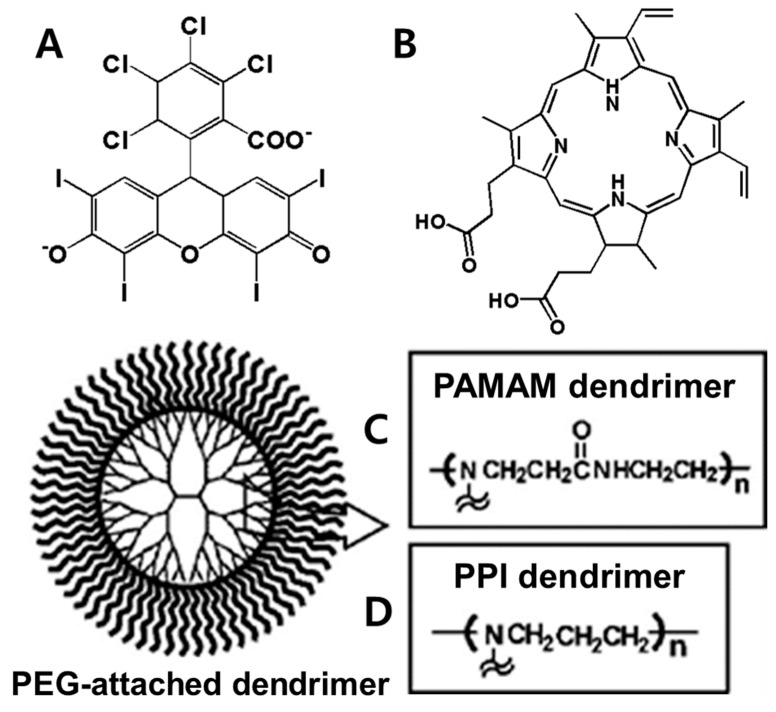
Structures of RB (**A**), PpIX (**B**), and PEG-attached PAMAM (**C**) and PPI (**D**) dendrimers. RB: rose bengal; PpIX: protoporphyrin IX; PEG: poly(ethylene glycol); PAMAM: poly(amido amine); PPI: poly(propylene imine). Reproduced with permission from Kojima, C.; Toi, Y.; Harada, A.; Kono, K. Preparation of poly (ethylene glycol)-attached dendrimers encapsulating photosensitizers for application to photodynamic therapy. *Bioconjug. Chem.*
**2007** [169].

**Figure 10 biomedicines-08-00618-f010:**
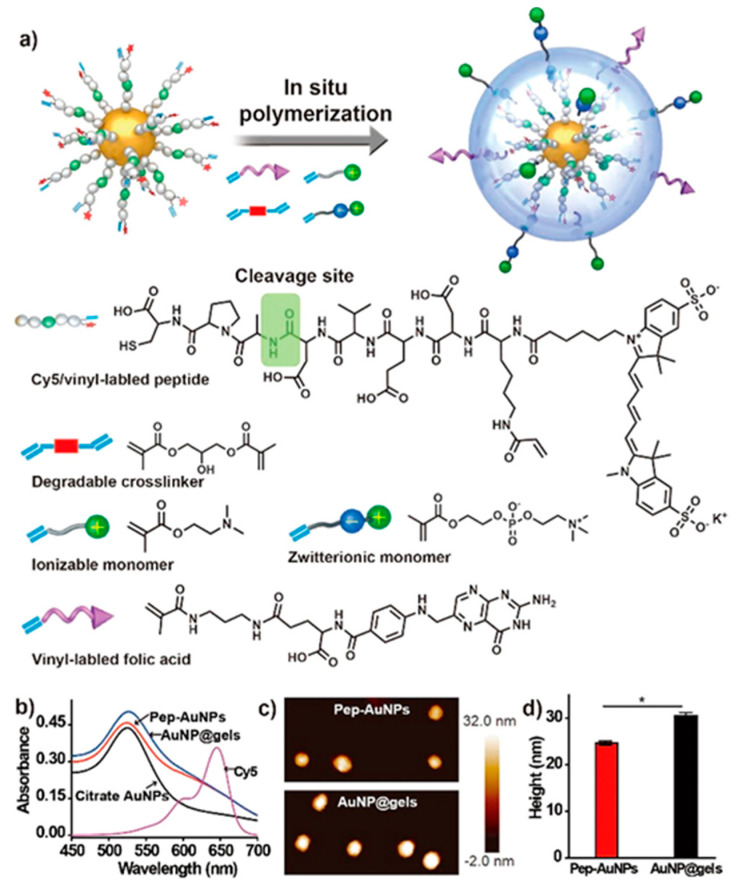
(**a**) Schematic illustration of the synthesis of a AuNP@gel probe through in situ polymerization of vinyl-bearing acid-degradable cross-linkers, ionizable monomers, zwitterion monomers, and cancer-cell-targeting vinyl-labeled folic acids on an AuNP core modified with Cy5/vinyl-labeled CPADVEDK peptides. The green frame in the peptide sequence indicates the cleavage site for caspase-3/-7. (b) UV/vis spectra of citrate-coated AuNPs, pep-AuNPs, AuNP@gels, and Cy5 dye solutions. (c) Atomic force microscopy (AFM) images of pep-AuNP (top) and AuNP@gel (bottom) probes. (d) Statistical sizes of pep-AuNP and AuNP@gel probes in part c (* *p* < 0.05). Reproduced with permission from Li, Q.; Qiao, X.; Wang, F.; Li, X.; Yang, J.; Liu, Y.; Shi, L.; Liu, D. Encapsulating a single nanoprobe in a multifunctional nanogel for high-fidelity imaging of caspase activity in vivo. *Anal. Chem.*
**2019** [176].

**Figure 11 biomedicines-08-00618-f011:**
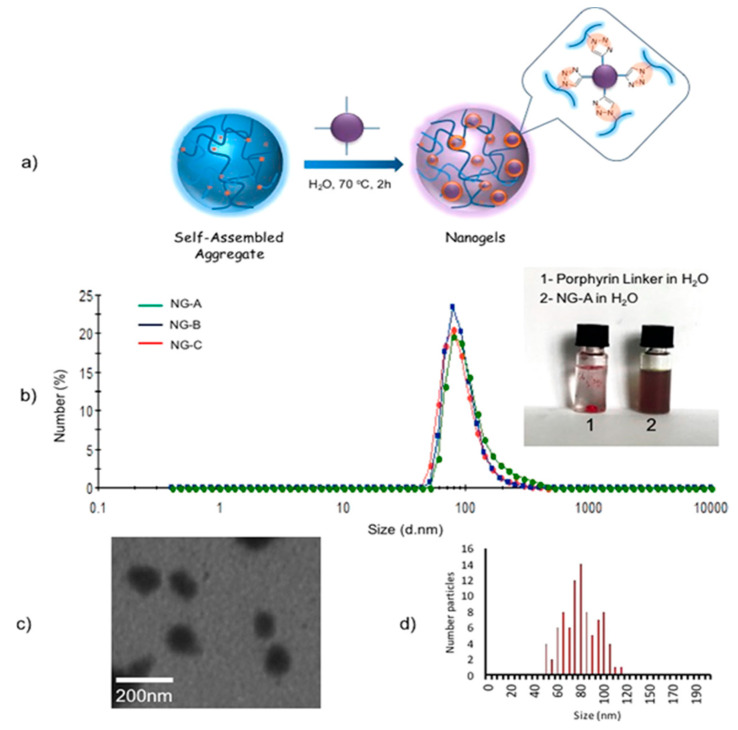
(**a**) Schematic representation of gelation reaction of copolymer poly(PEGMEMA-*co*-AHMA). (**b**) DLS data of three sets of nanogels (NG-A, NG-B, and NG-C) at 25 °C. (**c**) TEM image of nanogel NG-A. (**d**) Size distribution of NG-A from TEM analysis. Reproduced with permission from Chambre, L.; Saw, W.S.; Ekineker, G.; Kiew, L.V.; Chong, W.Y.; Lee, H.B.; Chung, L.Y.; Bretonnière, Y.; Dumoulin, F.; Sanyal, A. Surfactant-free direct access to porphyrin-cross-linked nanogels for photodynamic and photothermal therapy. *Bioconjug. Chem.*
**2018** [177].

**Table 1 biomedicines-08-00618-t001:** Benefits and risks of photo-based imaging and therapy.

Benefits	Ref.
High selectivity	[4,30]
High efficacy and low/no systemic toxicity	[4,30]
Light irradiation in the location of lesions can be controlled well	[4,13]
Minimally or non-invasive and effective modality	[13,30,31,32]
Convenient method	[13,30,31,32]
**Risks**	
Photic injury, photochemical injury, and photomechanical damage	[13,33]
Phototoxicity	[13,33]
Frequent PT treatments can lead to immunosuppression	[34]
Increased risk of developing skin cancer	[35,36]

PT: phototherapy.

**Table 2 biomedicines-08-00618-t002:** Imaging PSAs used in OI.

Type of PSAs	PSAs	Structure	Imaging Modality	Ref.
Fluorescent dye	Coumarin	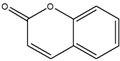	Fluorescence	[63]
Fluorescent dye	Fluorescein	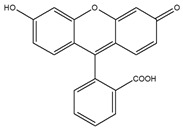	Fluorescence	[64]
Fluorescent dye	Alexa Fluor	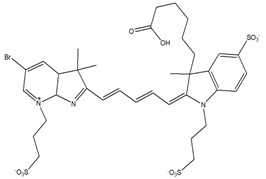	Fluorescence	[65]
Fluorescent dye	Cyanine	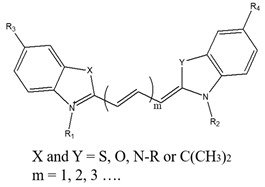	Fluorescence	[64,66]
Quantum dot	CdSe/Zns	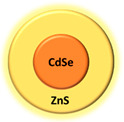	Fluorescence	[67]
Quantum dot	CdS_x_Se_1-x_/ZnS	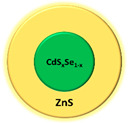	Fluorescence	[67]
Metallic NPs	Gold NPs and gold nanorods	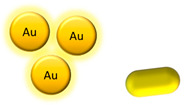 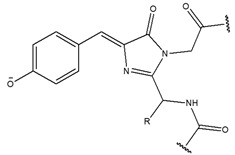	Fluorescence	[68,69]
Fluorescent protein	Green fluorescent protein (GFP)	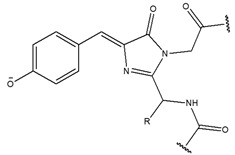	Fluorescence	[70]
Fluorescent protein	Red fluorescent protein (RFP)	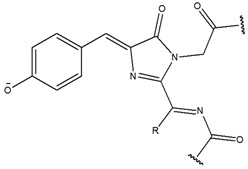	Fluorescence	[70]
Bioluminescent probe	Luciferin	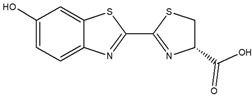	Bioluminescence	[71]
Organic compound	Porphyrin	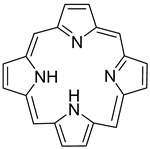	Fluorescence	[72]

PSAs: photosensitive agents; NPs: nanoparticles.

**Table 3 biomedicines-08-00618-t003:** Commonly investigated PT agents.

Type of PSAs	PSAs	Modality	Ref.
Tetrapyrrole	Porphyrin	PDT, PTT	[109,110]
	Chlorin	PDT	[111]
	Phthalocyanine	PDT, PTT	[112]
	Bacteriochlorin	PDT	[113]
Natural compound	Hypericin	PDT	[114]
	Hypocrellin	PDT	[115]
	Riboflavin	PDT	[116]
	Curcumin	PDT	[117]
Other photoactive dye	Methylene blue	PDT	[118]
	Toluidine blue	PDT	[119]
	Rose Bengal (RB)	PDT	[120]
	Squaraine	PDT	[121]
	Boron dipyrromethene	PDT	[122]
	Phenalenones	PDT	[123]
	Indocyanine green (ICG)	PDT, PTT	[124,125]
Inorganic NPs	Titanium dioxide (TiO_2_)	PDT	[126]
	Zinc oxide (ZnO)	PDT	[126]
Metallic NPs	Gold NPs	PDT, PTT	[102,127]
Carbon-based NPs	Fullerene	PDT	[126]
	Graphene	PDT, PTT	[126,128]
Quantum dots (QDs)	Ge-QDs, Ag_2_S QDs, CdS, CdSe, PbSe, InP, CdTe, and tungsten sulfide (WS_2_) QDs	PDT, PTT	[129]

PT: phototherapy; PSA: photosensitive agents; NPs: nanoparticles; PDT: photodynamic therapy; PTT: photothermal therapy.

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
