# Peer review of "Photo-Based Nanomedicines Using Polymeric Systems in the Field of Cancer Imaging and Therapy"

_biomedicines, 2020, doi:10.3390/biomedicines8120618_

Round 1
Reviewer 1 Report
This review article about integrating photochemistry based approach into nanomedicine for therapeutic and diagnostic purposes is well organized and well written. Authors have covered the utility of all the related sub-topics including photothermal, fluoresce and singlet oxygen generation and their integration into nanomedicine. Only few minor suggestion to improve the manuscript are as follows-
1) Micelles, which are monolayered, has been discussed but a closely related and very widely used multimodal drug delivery system, liposome has not been discussed. Liposomes are lipid bi-layers and deserves a separate section to be discussed. A related article is cited below-
Sadasivam M, Avci P, Gupta GK, Lakshmanan S, Chandran R, Huang YY, Kumar R, Hamblin MR. Self-assembled liposomal nanoparticles in photodynamic therapy. Eur J Nanomed. 2013 Jul;5(3):10.1515/ejnm-2013-0010. doi: 10.1515/ejnm-2013-0010. PMID: 24348377; PMCID: PMC3857307.
2) Several of the figures suffer from low resolution even though those are well explanatory. Especially the chemical structures in Figures 3a, 8 and 9 requires improvements.
Author Response
- Micelles, which are monolayered, has been discussed but a closely related and very widely used multimodal drug delivery system, liposome has not been discussed. Liposomes are lipid bi-layers and deserves a separate section to be discussed.
â–¶ Thank you for your suggestion. In this review, we address the benefits and risks of using light, imaging agents, phototherapy, and photosensitive nanomedicines derived using polymers of different architecture (page 2, line 59-61). Therefore, this review is focused on polymeric-based nanomedicine. On the other hand, liposomes are formed by one or more lipid bilayers. Considering your suggestion, we add sentence in page 2 line 57-59 as follows:
(page 2, line 57-59)
The most widely studied drug delivery system based on nanoparticle technology are liposomes, polymers and solid inorganic NPs [16]. Among them, we focus on polymeric-based nanomedicines.
- Several of the figures suffer from low resolution even though those are well explanatory. Especially the chemical structures in Figures 3a, 8 and 9 requires improvements.
â–¶ Appreciated for your valuable comments. The chemical structures in figure 3a, 8 and 9 was rearranged and improved with the resolution as the attachment.

Reviewer 2 Report
In this review, authors focused on the photosensitive agents as nanomedicines. This review was well organized and the inserted figures and table were also valuable. So this review was very useful to understand the nanoparticles and polymer-based nanomedicine. Finally I suggested this review was acceptable for the publication.
Author Response
â–¶ Thank you for your warm comments.

This manuscript is a resubmission of an earlier submission. The following is a list of the peer review reports and author responses from that submission.